# Seed Morphometry Reveals Two Major Groups in Spanish Grapevine Cultivars

**DOI:** 10.3390/plants14101522

**Published:** 2025-05-19

**Authors:** José Javier Martín-Gómez, José Luis Rodríguez-Lorenzo, Francisco Emanuel Espinosa-Roldán, Félix Cabello Sáenz de Santamaría, Gregorio Muñoz-Organero, Ángel Tocino, Emilio Cervantes

**Affiliations:** 1Instituto de Recursos Naturales y Agrobiología de Salamanca (IRNASA)—Consejo Superior de Investigaciones Científicas (CSIC), Cordel de Merinas, 40, 37008 Salamanca, Spain; jjavier.martin@irnasa.csic.es; 2Plant Developmental Genetics, Institute of Biophysics v.v.i, Academy of Sciences of the Czech Republic, Královopolská 135, 612 65 Brno, Czech Republic; rodriguez@ibp.cz; 3Instituto Madrileño de Investigación y Desarrollo Rural Agrario y Alimentario (IMIDRA), Finca El Encín, 28805 Alcalá de Henares, Spain; franciscoemmanuel.espinosa@madrid.org (F.E.E.-R.); felix.cabello@madrid.org (F.C.S.d.S.); gregorio.munoz@madrid.org (G.M.-O.); 4Departament of Matemathics, Faculty of Sciences, University of Salamanca, Plaza de la Merced 1–4, 37008 Salamanca, Spain; bacon@usal.es

**Keywords:** grape diversity, morphometric geometry, shape analysis, solidity, *Vitis* cultivars

## Abstract

Seed morphological description requires quantitative methods for further comparison. Here, traditional measurements, curvature analysis, and the *J*-index (percentage of similarity to a geometric model) were applied to the average contours (Acs) of 271 *Vitis* cultivars from the Spanish collection at IMIDRA (Madrid, Spain), including 9 different *Vitis* species and several sylvestris seeds (i.e., those derived from plants that once grew in the wild). Acs are graphical representations of the shape in seed populations, which can be obtained either from image analysis programs or computationally opening the way to quantitative analysis. A geometric model is a geometrically defined, closed curve, used as a reference for shape quantification. Based on existing differences between the Hebén cultivar (collected in 2020 and 2024; Hebén model, for morphotype 1) and the European varieties Chenin and Gewurztraminer (Chenin model, for morphotype 2), we created two models. The comparisons were based on a *J*-index, resulting in four groups: Group 1 contained all seeds with values lower than 90 for both models and included all *Vitis* species other than *V. vinifera* and most sylvestris seeds; Groups 2 and 3 contained seeds with *J*-index values higher than 94 for the Hebén and Chenin models, respectively. Group 4 consisted of seeds not included in the other groups. Based on *J*-index values, differences in curvature and solidity, and PCA analysis with Fourier coefficients, this work defines two new morphotypes associated with the Hebén (Group 2) and Chenin (Group 3) models, related to Iberian and Western European varieties, respectively.

## 1. Introduction

*Vitis vinifera* L. is one of the oldest species domesticated in agriculture [1,2,3]. Thousands of cultivars are now used in different countries around the world as a source of grapes for wine production or for direct consumption as fruit. In addition to the economic prosperity of the wine industry and the historical analysis of this culture, the number and diversity of cultivars represent an interesting paradigm for the analysis of biodiversity [3,4,5,6]. A small number of varieties account for a large proportion of cultivated grapes, such as the Spanish varieties Airén, Garnacha, and Tempranillo, or the French Cabernet Sauvignon, Chardonnay, and Merlot [6].

The characterisation of grape varieties is a goal of modern viticulture. This task includes identification, pedigree analysis, and parental relationships, and it is the basis for sustainable breeding and adaptation to the demands of changing environmental conditions. The description of *Vitis* cultivars is based on both physical characteristics of the plant (ampelography) and molecular markers. Thus, phylogenies have been based on leaf shape [7,8] as well as on DNA sequences [9,10,11,12,13,14]. Spanish cultivars have been classified based on nuclear microsatellite loci (SSRs) [15,16,17]. In some cases, parental relationships are known, either from biographical information or from genetic analysis, or both. For example, the vines Hebén (female) and Cayetana Blanca have numerous descendants among the Iberian cultivars, which has been confirmed by SNPs [18,19,20,21,22,23].

Seed morphometric analysis can be a useful approach for cultivar identification and characterisation, as well as for parental relationship analysis. Traditionally, seed morphometrics has been based on size measurements and their relationships. The Stummer index (SI) [24,25] is a measure often used in *Vitis* studies for germplasm and archaeological studies [26,27]. The SI is the inverse of the aspect ratio (ratio of length to width), a measure often used in general morphological analysis. Together with solidity, both are useful in describing the seeds of many species and cultivars. Solidity is the ratio of the area of any plane object to the area of its convex envelope and is the measure with the lowest rate of variation in species of *Vitis* and other genera [28,29].

The values of aspect ratio and solidity are useful shape descriptors for *Vitis* seeds, but the description is improved when supported by the visual representation of images representing the seed contour and the quantitative comparison with models. In previous studies, we described a method for obtaining the average contours (Acs) of seeds of different species and cultivars using image processing software [28,30], but this can also be done algebraically based on Fourier equations. Fourier analysis is a mathematical method that transforms closed curves into their component spatial frequencies. The parametric functions representing a closed plane curve can be approximated by sums of trigonometric functions and their truncated Fourier expansions [31,32]. The Elliptic Fourier Transform (EFT) has been applied to the shape analysis of *Vitis* seeds, finding a relationship between seed shape and taxonomic position [33], and to the analysis of archaeological seeds [34]. The coefficients corresponding to the Fourier equation defining the contour of the seed images can be obtained using the Momocs 1.4.0 software [35] in the R environment [36].

PCA based on the coefficients of the Fourier transform revealed differences between the seeds of *Vitis vinifera* cultivars and those of *V. vinifera* subsp. *sylvestris* [33]. In this work, seed shape was analysed in a total of 271 *Vitis* cultivars and species. These include eight species of *Vitis* other than *V. vinifera*, as well as seeds obtained from sylvestris plants, i.e., those plants maintained in the IMIDRA collection and derived from plants that once grew in the wild, and also cultivars common in the different geographical regions of Spain and Portugal (Airén, Castelao, Fernao Pires, Tempranillo, Verdejo, Viura), together with minority [37] and European cultivars (Chasselas Blanc, Gewürtztraminer, Pinot Noir). After a general morphometric analysis including shape measurements (area, perimeter, length, width, circularity, aspect ratio, roundness, and solidity), eight Fourier coefficients were obtained with Momocs for samples of 20–30 seeds representing each of the 271 cultivars analysed, and the average contour was plotted for each cultivar. The contours of all varieties were numbered based on aspect ratio and solidity. For shape comparisons, two new models were defined based on the contours of morphologically different and reference cultivars. Model 1 was made with the average contour of the seeds of the cultivar Hebén (2020 and 2024; Hebén model, for morphotype 1) and model 2 with an average contour of the European varieties Chenin and Gewürtztraminer (Chenin model, for morphotype 2). The calculation of *J*-index (percent of similarity) of 271 cultivars with these two models resulted in four groups. Group 1 contained all seeds giving values lower than 90 with both models and included all *Vitis* species other than *V. vinifera* and most sylvestris seeds; Groups 2 and 3 contained seeds with *J*-index values higher than 94 with the Hebén and Chenin models, respectively. Group 4 consisted of seeds not included in the other groups. Group 2 contained mainly cultivars related to Hebén, while Group 3 contained seeds of different origin. The average solidity was higher in Group 2 than in Group 3, and differences in curvature allowed us to further define morphotypes corresponding to models Hebén and Chenin.

## 2. Results

### 2.1. Average Contours for Each Cultivar and Their Validation

Fourier equations with eight coefficients were obtained for each seed with Momocs, and the average contours (Acs) were plotted for each of the 271 cultivars or species. Both the original seed images and the Acs are available (see Appendix A). Acs were validated by two independent methods: (1) similarity with Acs derived from image analysis (Figure 1) and (2) calculation of *J*-index (percent similarity) with the original seeds from which they were derived (Table 1).

### 2.2. A Preliminary Classification of Cultivars by Aspect Ratio and Solidity

The average aspect ratio values for the seeds of 271 cultivars ranged from 1.28 to 2.40, which formed the basis for the division into two groups of high and low aspect ratio, with values between 1.58 and 2.40 in the seeds of the first group (136 cultivars) and between 1.28 and 1.57 in the second group (135 cultivars). Images are available for all Acs numbered according to this classification (see Appendix A). The boundary between the two groups corresponds to a Stummer index intermediate between those of wild and cultivated seeds [1,24]. In agreement with this, the seeds of *Vitis* species other than *V. vinifera* and the majority of sylvestris populations were included in the lower aspect ratio group. Within each of the two groups, the cultivars were numbered from 1 (Dominga) to 136 (de Cuerno) in the first group, and from 137 (*Vitis candicans*) to 271 (Coloraillo) in the second, in descending order of seed solidity. Thus, among the varieties with a high aspect ratio, the seeds of Dominga had the highest solidity (967), and those of de Cuerno the lowest (909), while in the group with a low aspect ratio, *V. candicans* had the highest solidity (980) and Coloraillo the lowest (921). Low aspect ratio and high solidity characterised the seeds of *Vitis* species other than *V. vinifera* and those of sylvestris plants.

### 2.3. Two New Models

Two new models were developed algebraically. The Hebén model was the graphical representation of the Fourier equation with the mean coefficients of the Acs of Hebén 2020 and Hebén 2024. The Chenin model was made with the mean of the Acs of the varieties Chenin and Gewürztraminer. Chenin is a descendant of Savagnin, and Gewürztraminer is a mutation of this cultivar. Both models were designed on the basis of their morphological peculiarities (higher solidity in Hebén, lower in Chenin) as well as their potential representativeness for the Iberian and West European varieties. They are shown in Figure 2 together with the As of Hebén 2020 and 2024 (superimposed on the Model Hebén), and with Chenin and Gewürztraminer (superimposed on the Model Chenin).

### 2.4. Classification of 271 Varieties Based on the Similarities of the Average Contours of Their Seeds with Two Models

*J*-index values were calculated for the Acs of the seeds of 271 cultivars with each of the two new models (Appendix A). The results ranged from 73.9 (de Cuerno) to 97.2 (Merseguera 2020) with the Hebén model and from 68.3 (de Cuerno) to 97.0 (Tinto de la Pampana Blanca) with the Chenin model. The 271 varieties were divided into four groups according to their *J*-index values: Group 1: 69 cultivars with *J*-index values lower than 90 with both models; Group 2: 60 cultivars with *J*-index values higher than 94 with the Hebén model; Group 3: 44 cultivars with *J*-index values higher than 94 with the Chenin model; and Group 4: 98 cultivars with *J*-index values comprised between 90 and 94 with either model and not included in any of the previous groups. Group 4 was divided into two subgroups: 4a, of 58 cultivars having higher values with Hebén than Chenin, and 4b, of 40 cultivars with higher *J*-index values with Chenin than with Hebén.

Of the 69 seed contours in Group 1 (Figure 3), eight belong to different species of *Vitis* other than *V. vinifera*; 13 belong to sylvestris populations; two belong to Tortozona Tinta in two different years (2020 and 2024); and, of the remaining 46 cultivars, 25 are of unknown origin. The remaining 21 are the result of diverse crosses, including some with Hebén (Alarije, Jerónimo, Moscatel de Angués, Pardillo, Perruno, Tarragoní) or closely related to the Chenin model varieties such as Albarín Blanco, Blanquiliña, Lado, and Parduca. These all result from crosses with Savagnin Blanc [18]. This group contains cultivars from at least 17 different crosses.

### 2.5. Two Groups Defined by the Models Differ in the Shape of Their Seeds

Figure 4 shows the contours of 60 cultivars that make up Group 2. Of them, 21 are the progeny of Hebén (Merseguera, Xarello, Trepat, Verdejo de Salamanca, Quigat, Señá, Sabro, Alcañón, Esperó de Gall…), while two are the progeny of Hebén’s sons, Ferral’s (Don Mariano) and Beba’s (Tortosí). One of the contours in this group belongs to a sylvestris population, and 21 are from unknown crosses. The remaining 15 proceed from diverse crosses: three from Savagning blanc (Verdejo is a progeny of Castellana Blanca and Savagnin Blanc, and Pinot Noir and Cagarrizo of Savagnin Blanc with unknown cultivars); Fernao Pires is a progeny of Malvasia Fina [18]. Others descend of crosses of Castellana Blanca with Breval negro (Quiebratinajas Rosa), or with unknown cultivars (Moraté).

Chenin Blanc and Gewürztraminer are both linked to Savagnin. However, Chenin Blanc is the offspring of Savagnin Blanc, while Gewürztraminer is a mutation of this variety. The contours of both were selected based on their similarity (low solidity). Group 3 of contours, defined by their high percentage of similarity to this model made as the average contours of Chenin Blanc and Gewürztraminer, includes three cultivars obtained in crosses with Savagnin Blanc: Prieto Picudo and Sauvignon Blanc proceed from crosses with unknown cultivars and Maturana Blanca from the cross between Savagnin Blanc and Castellana Blanca. Some of the cultivars proceed from Hebén in crosses with unknown varieties, or with Breval Negro (Torralba), Brustiano Faux (Xarello Rosado), Graciano (Mandón), or Moscatel de Alejandría (Moscatel de Angués) [18] (Figure 5). This group includes also cultivars from 16 crosses of unknown cultivars and others involving diverse parentals.

Group 4a of 58 contours, defined by relatively high percentage of similarity to the Hebén model, includes cultivars related to the Hebén genealogy, such as Mantúo de Pilas, Macabeo, Miguel de Arco, Verdil, Cayetana Blanca, Zalema, Ferral, Sumoll, Gorgollasa, Cadrete, Planta Fina, Corazón de Cabrito, or Malvar [18] (Figure A1 in Appendix B). There are also cultivars from crosses of Cayetana Blanca, a descendent of Hebén (Juan García, Jaén Tinto, Rocía), as well as the progeny of diverse crosses of Castellana Blanca (Cariñena Blanca, Morenillo) or Savagnin Blanc (Carrasquín, Alfrocheiro, Merenzao, Gewürztraminer). Five contours correspond to sylvestris, and eighteen are the progeny from crosses involving unknown and others with diverse cultivars.

Group 4b of contours, defined by relatively high percentage of similarity to the Chenin model, includes some cultivars related to the Savagnin Blanc/Chenin genealogy, such as Gewürtztraminer and Merenzao (syn. Trousseau Noir), but also other descendents of Hebén (Vijariego Blanco, Planta Fina, Negreda, Moscatel de Angués), or descendents from both genealogies (Trincadeira das Pratas) [18] (Figure A2 in Appendix B).

### 2.6. Comparison Between Two Groups Reveals Difference in J-Index, Aspect Ratio, and Solidity

The comparison between Groups 2 and 3 was based on the following measurements: *J*-index of their Acs with both models, aspect ratio and solidity. The results are shown in Table 2.

### 2.7. Curvature Values Contribute to Define Two Morphotypes

Curvature values were measured in the contours on which the models were based and four additional contours representative of each group (Figure 6). Figure 7 presents two samples of curvature analysis in contours of Hebén 2024 and Gewürtztraminer. Differences were found for the *J*-index with the model Hebén, curvature and solidity (Table 3).

### 2.8. PCA Defines Three Morphological Types

The results of factorial analysis (PCA) with the Fourier coefficients are shown in Figure 8. PCA is based on the coefficients of parametric Fourier equations: A1 to A8, and B1 to B8 are, respectively, the cosine and sine coefficients for the first eight harmonics corresponding to the x-coordinate. C1 to C8, and D1 to D8, are, respectively, the cosine and sine coefficients for the first eight harmonics corresponding to the y-coordinate. PC1 was integrated by coefficients A7, D2, and D5 and negatively by D4, D6, A3, and D1, and it accounted for 19.3% of the variation. PC2 was formed by B4, B6, and C6 and accounted for 14.3% of the variation. The graphic representation of the values of PC1 and PC2 for 59 cultivars of Group 1, 49 of Group 2, and 29 of Group 3 is shown in Figure 8. The centroid of Group 2 (Hebén, red) is located between those of Group 3 (Chenin blue) and Group 1 (sylvestris and *Vitis* species other than *V. vinifera*, green).

## 3. Discussion

Recent research in *Vitis* offers new perspectives on general aspects of plant biology, from the determination of sexual types in response to environmental conditions [38] to genomic changes resulting from different reproductive mechanisms [39]. Seed morphology has been the subject of intense research [1,24,25,26,27,28,29,30], with the aim of identifying genetic and epigenetic factors that determine seed shape, for which accurate morphological descriptions are required [30,33,34,40,41]. The reproductive systems of *Vitis* alternate between sexual reproduction with dioecy in natural populations and vegetative propagation with eventual crosses in hermaphroditic plants in culture. New cultivars have been formed by sexual reproduction in crosses, sometimes hundreds of years ago, and have maintained their genetic characteristics through subsequent processes of vegetative propagation. As in the analysis of progeny by genetic markers, where stability is an important requirement, morphological analysis is based on the stability of seed shape between generations of vegetative propagation. In each generation, both autogamy by pollen from the same plant or other plants of the same variety and cross-pollination by plants of other varieties in the neighbourhood can occur. In the case of female plants, the second alternative is obligatory, as no cases of apomixis have been reported in Vitaceae [42]. Pollination by plants of other varieties may affect seed shape in one generation, but it does not change the genetic composition of the vegetatively propagated plant, and consequently the female genotype predominates over generations.

Hebén is a female cultivar known to be the mother of dozens of cultivars in the Iberian Peninsula [18,21,22,23,40]. Recently, we investigated the conservation of seed shape between Hebén and some of the cultivars in her progeny [40]. Seed shape was relatively conserved for most of the cultivars in the progeny of Hebén, and the values of the *J*-index with a model derived from Hebén in the seeds of Forcallat Tinta, Merseguera, and Verdejo de Salamanca were similar to those obtained in the seeds of Hebén itself [40].

In this work, the average contours (Acs) of 271 populations of *Vitis*, including eight belonging to different species of *Vitis* other than *V. vinifera* and 21 seed populations of *Vitis vinifera* subsp. *sylvestris,* have been compared with two new models. Both the Acs and the models were based on the application of the Momocs programme [35] to images containing 20 or 30 seeds of each genotype. This provided the coefficients for eight harmonics of the Fourier equations representing the seeds in each image, which were plotted using Mathematica. The new models were designed as Hebén and Chenin, each representing the Acs of two populations: Hebén is the average of Acs of Hebén seeds harvested in 2020 and 2024, and Chenin is the average of Chenin and Gewurztraminer seed contours. The models were selected based on observed differences in the shape of their seeds in aspect ratio and solidity. Their aspect ratios are of 1.62 (Hebén) and 1.68 (Chenin model), and both are in the range of the Stummer index, which is considered to belong to seeds of cultivated varieties and not to sylvestris plants [1,24]. Solidity is higher in Hebén. The comparison of these two models with the Acs of 271 cultivars resulted in four groups. Group 1 consisted of 69 cultivars with *J*-index values below 90 with both models. It included most sylvestris genotypes and all seeds belonging to different species of *Vitis* other than *V. vinifera* [28], in addition to 25 cultivars of unknown origin and 21 resulting of diverse crosses, including some with Hebén (Alarije, Jerónimo, Moscatel de Angués, Pardillo, Perruno, Tarragoní) or closely related to the Chenin model varieties, such as Albarín Blanco, Blanquiliña, Chenin, Lado, and Parduca. These all result from crosses with Savagnin Blanc [18]. Related to the sylvestris genotypes, they can either belong to *V. vinifera* subsp. *sylvestris* or be feral plants, i.e., have escaped from cultivation. The choice between these alternatives is complex and may be based on the aspect ratio values, with low aspect ratios corresponding to *V. vinifera* subsp. *sylvestris* and high aspect ratios to feral seeds. In addition, the method based on the PCA of the Fourier coefficients has revealed differences between wild grape accessions and cultivated varieties [33], and this was confirmed here by the differences found between Group 1 and Groups 2 and 3 in the PCA.

It was not surprising to find that some of the varieties related to the models had low *J*-index values with them. As observed in the analysis of the progeny of Hebén [40], variations in aspect ratio are notable among the varieties of a progeny, and these can strongly influence the values of *J*-index. Other reasons for the different values of *J*-index between a parental and the progeny may be related to the presence of a dominant component in the other parental not related to the model.

The values of the *J*-index in this work are higher than those reported previously [40], because here both the model and the contours to be analysed were derived from Fourier equations, providing smoother curves than the contours of the seed images. The superposition of two smooth curves has a high degree of coincidence compared to complex curves rich in non-convex regions of negative curvature.

In addition to the progeny of Hebén analysed in our previous work [40], Group 2 contained more descendants of Hebén with a similarity of their seeds to the Hebén model, among them Alcañón, Esperó de Gall, Palote, Parellada, and Planta Fina. Also, Garrido Fino is related to Hebén, as it is the descendent Garrido Macho, itself a progeny of Cayetana Blanca [18,19,22,43]; Don Mariano and Tortosí are also descendants of Hebén, being the progeny of Ferral and Beba, respectively. Quiebratinajas and Verdejo are the progeny of Castellana Blanca in crosses with Breval Negro and Savagnin Blanc, respectively [43], and Moristel is the progeny of Moraté, a descendent of Castellana Blanca [43], suggesting that Castellana Blanca could be related to Hebén. Due to the high value of the *J*-index found in the comparisons of their contours with the model, Hebén or a related variety could be in the ancestry of some of the other varieties derived from unknown parents (Aledo Real, Espadeiro, Ferrón, Moscatel de Grano Menudo, Ondarrabi Beltza, Picapoll, Planta Mula, Rosetti, Rubeliza, Sousón, Terriza, Tinto Velasco, Tortozón, Treixadura) or from crosses whose parents are not known to be related to Hebén (Forastera, Manto Negro, Loureira).

Hebén and its descendants form a large and complex family tree, characterised by several molecular markers and, in many of them, also by the shape of their seeds, whose contours tend towards a triangular shape with high solidity. In a preliminary classification of cultivars based on the aspect ratio of the seeds, we observed that, in contrast to the Hebén-related cultivars, there was a group of seeds with a straighter peduncle (beak), as if the peduncle were delimited by two parallel lines. In accordance with this shape and the corresponding lower values of solidity, the Chenin model was made as an average of the mean contours of Chenin and Gewurztraminer. Of the total Acs, 60 gave values of *J*-index higher than 94 with this model. As in the case of the varieties related to Hebén above, some of the varieties with high *J*-index values with the Chenin model were of unknown origin (Vidadillo, Garnacha, Caíño Longo, Cuatendrá, Godello, Batista, Ratiño) or derived from crosses with varieties of partly known origin such as Forastera, from Biancolella or Loureira, from Amaral and Branco Escola [43]. Prieto Picudo and Sauvignon Blanc are the progeny of Savagnin Blanc; Casteloa is the descendent in a cross between Alfrocheiro (progeny of Savagnin Blanc) and Cayetana Blanca [44]. Also, some of the cultivars in this group were descendants of Hebén (Airén, Albillo Mayor, Cayetana Blanca, Derechero, Forcallat Tinta, Mandón, Mollar Cano, Montúa, Moscatel de Angués, Pedro Ximénez, Quigat, Sumoll, Torralba and Xarello) or Cayetana Blanca (Casteloa, Puerto Alto). The high percentage of descendents of Hebén found to have high *J*-index values with the Chenin Model may be explained because their male parentals could be related to the European cultivars.

The comparisons of percent similarity by *J*-index allow the identification of related cultivars. For example, Casteloa (descendent of Cayetana Blanca) and Morisca, a synonym of Cayetana Blanca, came close to each other in the classification by *J*-index values.

Differences between Groups 2 and 3, resulting of the quantitative comparison with models (*J*-index values), were found also in aspect ratio and solidity. Representative cultivars from each of these groups were selected for their curvature analysis defining two morphotypes, corresponding, respectively, to Hebén and cultivars of its progeny, as well as Chenin and related cultivars. Higher solidity is associated with lower curvature in the Hebén morphotype, while curvature values are higher in the Chenin morphotype, associated with lower solidity. Interestingly these two morphotypes may correspond to CG5 and CG6, two of the six groups defined as having pure or close to pure ancestries, which evolved in the Iberian Peninsula and Western Europe, respectively [45]. Among the cultivars analysed in this work, there are those representatives of the other genotypes described by Dong et al. [45], for example, Regina dei Vigneti and Muscat Hambourg of CG3, or Teta de Vaca (syn. Ahmeur bou Ahmeur), a mixture of CG1 and CG6 [45]. The detailed morphological analysis of more cultivars may reveal if the other groups differ also in the properties of their seeds, opening the way to the genetic definition of the elements controlling seed shape.

The results of the PCA based on the Fourier coefficients confirm the differences between Groups 1–3, based on the classification by *J*-index with the different models. Thus, in addition to the *V. vinifera* subsp. *sylvestris* morphotype, two new morphological types are defined by the *J*-index, solidity and curvature values, as well as by the differences observed in PCA with the Fourier transform coefficients.

The combination of quantitative measurements of shape, both traditional (aspect ratio, solidity) and others more recently introduced (*J*-index, curvature), will contribute to the definition of seed morphotypes, which is a prerequisite for the identification of genetic factors controlling seed shape. Recent developments in association mapping open new opportunities to explore the genomic regions associated with seed morphotypes, a work based on the identification of phenotypes and the accurate selection of parents in crosses [46].

## 4. Materials and Methods

### 4.1. Plant Material

The seeds in this work belong to 271 genotypes (Table 4). Most of them were harvested in 2020 and are described in Espinosa-Roldán et al. [41], 22 correspond to the cultivars harvested in 2024, as described by Cervantes et al. [40], and 9 correspond to *Vitis* species other than *V. vinifera* [28]. The “sylvestris” samples were 21 female plants surveyed between 2003 and 2009 in natural populations of riparian forests in the provinces of Badajoz, Caceres, Cadiz, Huelva, Jaén, Seville, Cordoba, Navarra, and the French Basque Country. These plants were selected for being female and therefore producing seeds, for being populations not contaminated with cultivated vines, and for representing samples from the entire Iberian Peninsula. Sylvestris plants can either belong to *V. vinifera* subsp. *sylvestris* or be ferals (plants from cultivars that have escaped cultivation).

### 4.2. Photography

Images were taken at IMIDRA (Madrid, Spain) for the seeds collected in August 2020 and at IRNASA (Salamanca, Spain) for the remaining seeds. In both cases, f/18 was the F-stop used, with a lighting between 800 and 1200 lumen and colour temperature of 6400 K. No filters were used to clean up images. A Nikon D80 10.2 megapixel camera (Nikon, Tokyo, Japan) equipped with a COSINA 100 mm f 3.5 MC Macro AF (Cosina Co., Ltd., Nakano, Japan), with an ISO of 400, was used and object distance of 43 cm in Madrid, as well as a Sony α 5100 24 megapixel camera (Sony, Tokyo, Japan) with an objective AF-S Micro NIKKOR 60 mm f/2.8 G ED (Nikon, Tokyo, Japan) with an ISO of 100 and object distance of 17 cm in Salamanca for the seeds of 2024. Shutter speed was manually set to S= 1/5 sec and the aperture to F = 16. In both cases, the ISO values were set manually. Focal distance of the objectives guarantees minimal distortion. In addition, the photographs of the seeds occupied the central part of the image, leaving an external part unoccupied to avoid distortions. Colour correction is not required because the images were transformed into the corresponding black contours. Image quality is enough (ISO values equal or lower than 400 guarantee that digital filters to reduce noise are not required). The seed images are available at Zenodo (see Appendix A).

### 4.3. Morphological Measurements

Measurements of seed images (area, perimeter, length, width, circularity, aspect ratio, roundness, and solidity) were completed with Image J 1.54 h [47]. The program converts pixels into mm according to a ruler contained in the images. For this, the cursor is placed on the start of the ruler, and a line is drawn corresponding to the ruler with the “Analyze”, “Set Scale” function. For example, the average area for 30 seeds of Hebén (2020) is 42,320 pixel (21 mm^2^). After adjusting the pixel-to-mm ratio, the images were converted to 8-bit, and the threshold was adjusted before the measurements (“Analyze particles”). Circularity, aspect ratio, roundness, and solidity are described in detail elsewhere [48,49,50]. Circularity is the following ratio:*C* = (4*π* × *A*)/*P*^2(1)
where *A* is area, and *P* is perimeter. Aspect ratio is the quotient L/W, where *L* is the length and W the width. Roundness is determined by*R* = (4 × *A*)/(*π* × *L*^2)(2)

Solidity is a property of closed-plane curves related to their convexity. It expresses the ratio of the area of an object to the area of its convex Hull (the convex Hull is the smallest convex set that contains a plane figure). Solidity is here given ×1000.

### 4.4. Extraction of Fourier Coefficients from the Images in Momocs

Images containing 20 or 30 seeds were converted into TPS files and the coordinates for each seed contour extracted following the protocol in Momocs [35], and the corresponding coefficients were applied to Mathematica equations to provide the average contour relative to each cultivar. The Acs for all the cultivars and species are available in Zenodo (see Appendix A).

### 4.5. Similarity Between Seed Contours and the Models: J-Index

The *J*-index is the percent of similarity between the seed image and a geometric model, i.e., a geometric figure taken as a reference. It is calculated by comparing both images. For this, the models are superimposed on images of 20 seeds of each cultivar, looking for maximum similarity. The models are overlaid on Corel PhotoPaint 24.5.0.731 (Corel Corporation; Ottawa, ON, Canada) containing 20/30 seeds of each cultivar, and two new files are saved for each of these images: one with the model in black and one with the model in white. The Image J program [47] gives the values of total area (T, being the contour of the model in black, the whole area is considered) and area shared between the model and the seeds (S, being the contour of the model in white, where the measured area is limited to the area shared between the seed and the model; Figure 9). Note that “T” is the total surface occupied by either the seed or the model, whereas in “S” is the measured surface is shortened by the white profile of the model. For each seed, the *J*-index is calculated as the ratio S/T. For each variety, the *J*-index is the mean value of its 20 (or 30) seeds. The images used for *J*-index calculation are available at Zenodo (see the Appendix A).

### 4.6. Curvature Analysis

Curvature analysis of seed silhouettes involved seed images of 150 ppp and resolved maximum, minimum, and average curvatures for the lateral view of each seed. A series of points delimiting the profile of the seeds was automatically taken with the function Analyze Line Graph of Image J. The corresponding Bézier curve and curvature measurements were obtained according to published protocols [51]. The curvature is provided in mm^−1^; thus, a curvature of 1 corresponds to a circumference of 1 mm, and a curvature of 10 to a circumference of 100 microns (0.1 mm). Maximum and minimum curvature values indicate the major changes in the slope in the curve. In a straight line, curvature equals 0, and in a circumference, curvature is constant. Thus, the higher the difference between maximum and minimum curvature values from the average curvature, the higher the departure from circularity. A low maximum to average curvature ratio indicates more constant curvature values related with a figure approaching a circumference. Negative values indicate changes in the direction of the slope and thus non-convex regions. Representative Mathematica files for curvature analysis are available at Zenodo (see Appendix A).

### 4.7. Statistical Analysis

As the populations did not follow a normal distribution, the comparisons were made using the Mann–Whitney U test. Statistical analysis was performed with IBM SPSS statistics v29 (SPSS 2022), with the level of significance *p* = 0.05. PCA was conducted with R [36].

## 5. Conclusions

Average contours (Acs) derived by the Morphometric software Momocs 1.4.0 well represent the shape of seed populations and are equivalent to image-program-derived Acs. Acs for 271 cultivars in the Spanish collection at IMIDRA were ordered and compared, with models derived from Hebén and Chenin/Gewürtztraminer revealing two different morphotypes that differ in curvature and solidity values. We hypothesise that these two morphotypes may correspond to genotypes CG5 and CG6, two of the six groups defined as having pure or close to pure ancestries, that evolved in Western Europe and the Iberian Peninsula, respectively. Seed morphological analysis based on *J*-index (percent similarity with a model), curvature, and solidity values may contribute to the description of major population events in the history of grapevine cultivation, as well as to provide new elements for the genetic analysis of development in *Vitis*.

## Figures and Tables

**Figure 1 plants-14-01522-f001:**
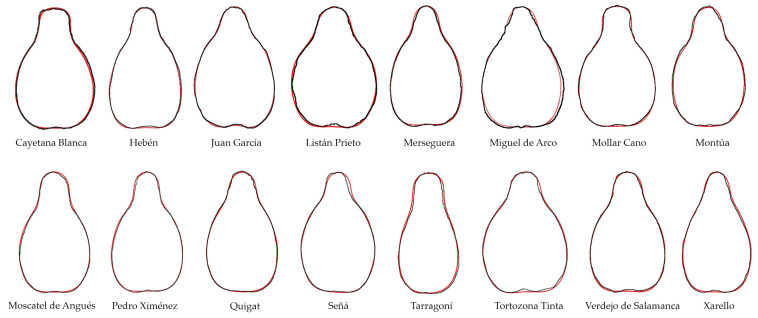
The curves representing Fourier equations with 8 coefficients obtained in Momocs (red) closely fitting the average contours resulting from the image analysis for each variety (black). Note that the red curves have a maximum of three regions of negative curvature (non-convex regions) corresponding to the laterals and the basis of the figures, while the black curves may have more non-convexity regions due to sinuosities. All the contours here correspond to seeds harvested in 2024.

**Figure 2 plants-14-01522-f002:**
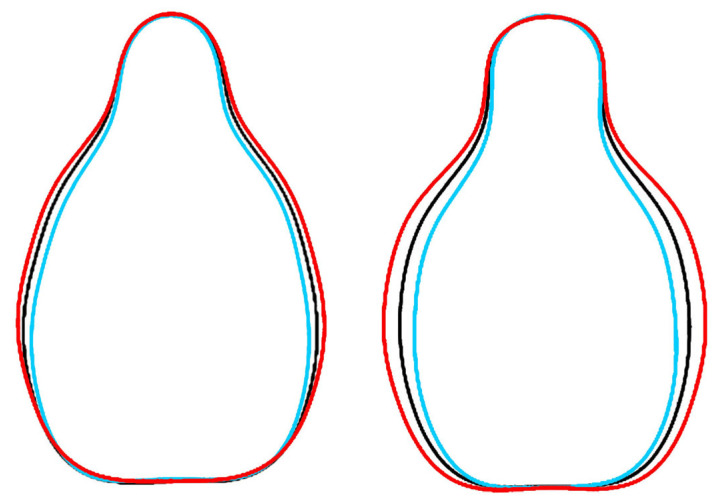
Models resulting from the representation of Fourier equations. (**Left**): Model Hebén (black) is the average of the mean contours of 30 seeds of Hebén 2020 (red) and 20 seeds of Hebén 2024 (light blue). (**Right**): Model Chenin (black) is the average of 20 seeds of Gewürtztraminer (red) and 20 seeds of Chenin (light blue). Values of aspect ratio are of 1.62 and 1.68, and solidity of 975 and 955, respectively, for the Hebén and Chenin models.

**Figure 3 plants-14-01522-f003:**
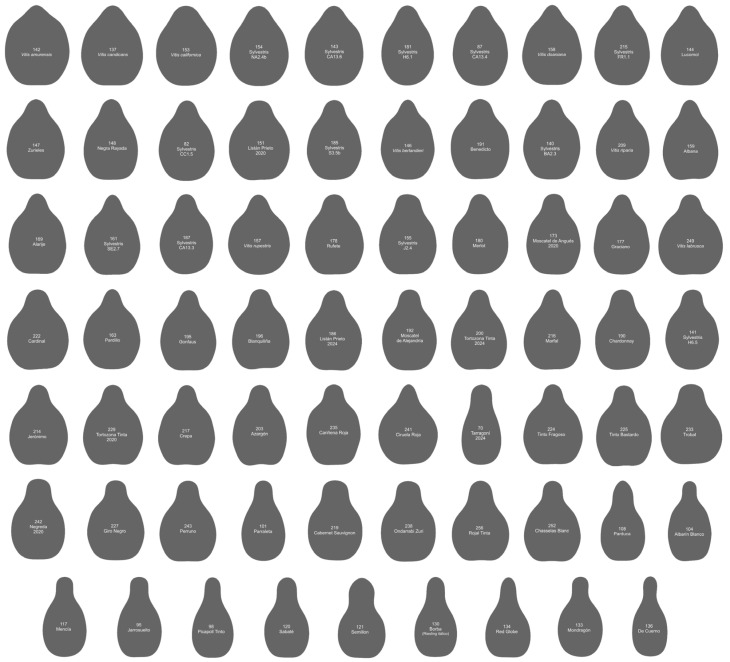
Average contours of 69 varieties with *J*-index values below 90 for the two models Hebén and Chenin. The numbers and names of the varieties are given within each contour. Cultivars are ordered from higher to lower solidity. Numbers 1 to 136 correspond to seeds of high aspect ratio; 137 to 271, low aspect ratio. Both series are ordered by decreasing solidity. The species and cultivars represented here are those of lower similarity to the models and include all seeds belonging to different *Vitis* species other than *V. vinifera*.

**Figure 4 plants-14-01522-f004:**
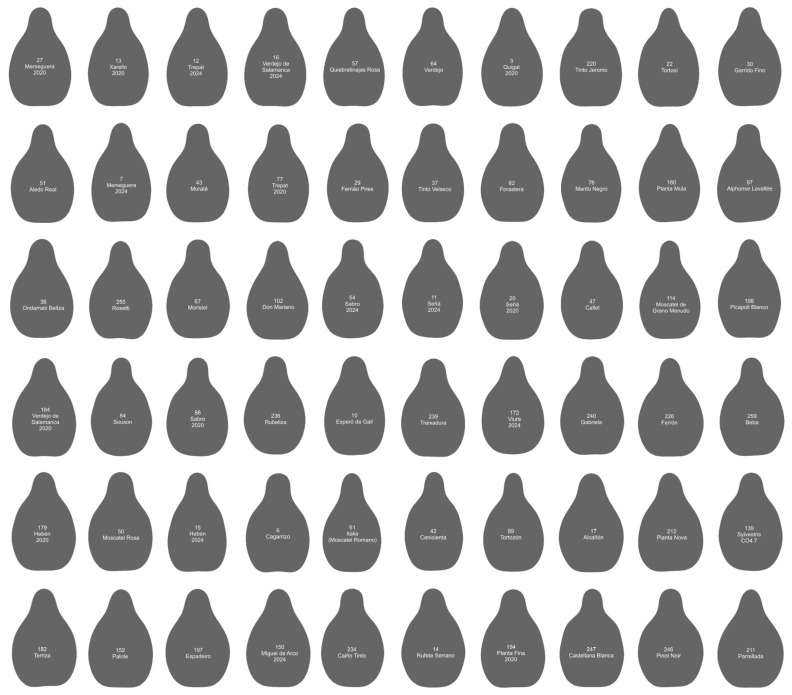
Contours of 60 cultivars giving *J*-index values superior to 94 with the model Hebén. The numbers and names of the varieties are indicated inside each contour. Cultivars are ordered from higher to lower *J*-index (percent similarity to Hebén model). The seeds more like the model Hebén are in the upper lines of the image.

**Figure 5 plants-14-01522-f005:**
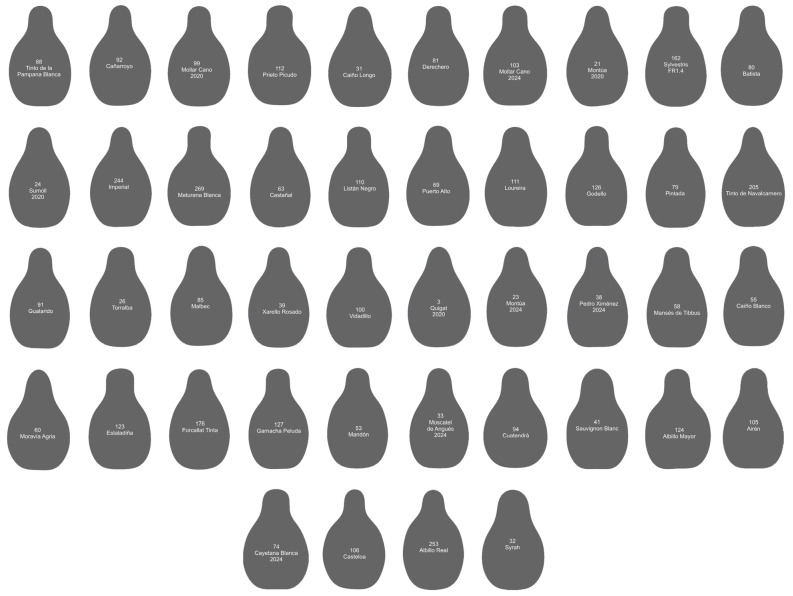
Contours of 44 cultivars giving *J*-index values superior to 94 with the model Chenin. The numbers and names of the varieties are indicated inside each contour. Cultivars are ordered from higher to lower *J*-index (percent similarity to Chenin model). The seeds more similar to the model Chenin are in the upper lines of the image.

**Figure 6 plants-14-01522-f006:**
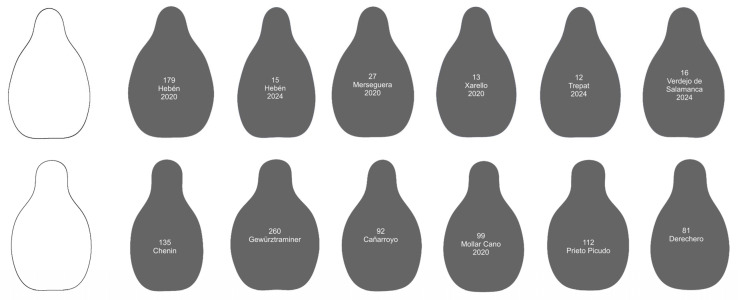
Contours representative of two morphotypes: Hebén (**above**) and Chenin (**below**), with their respective models (**left**).

**Figure 7 plants-14-01522-f007:**
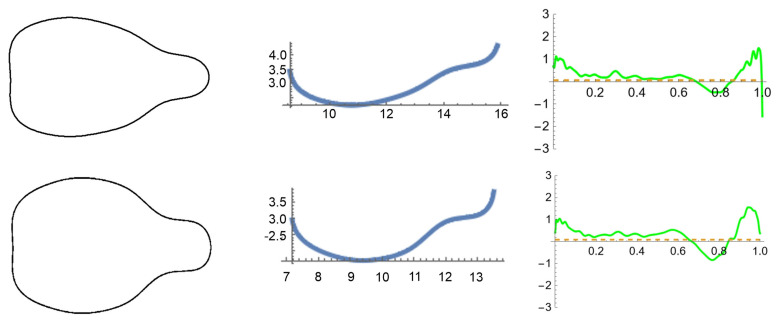
Curvature analysis in the lateral view of Hebén 2024 and Gewürtztraminer. (**Left**): Seed contours; (**Center**): Bezier curve corresponding to lower part of seed contours used to calculate curvatures; (**Right**): curvature values. Maximum, minimum, and average values were estimated between 0.2 and 0.8 to avoid conflicting regions in the extremes of the curve.

**Figure 8 plants-14-01522-f008:**
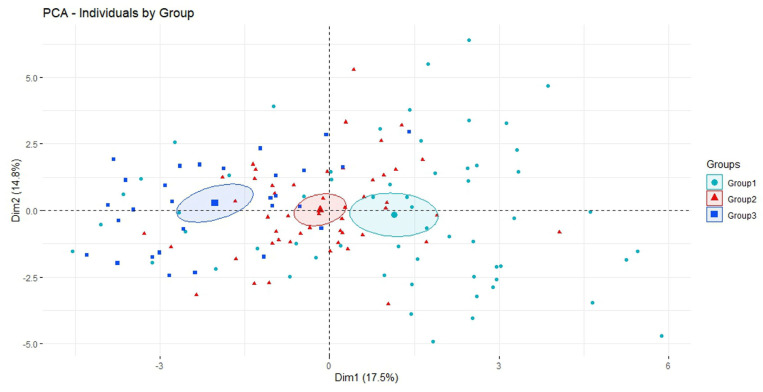
Results of PCA. Green points: 59 cultivars of Group 1, formed mainly by *Vitis* species and sylvestris plants. Red points: 49 cultivars of Group 2 (Hebén morphotype). Blue points: 29 cultivars of Group 3 (Chenin morphotype). Green square: the centroid for Group 1. Red triangle: the centroid for Group 2. Blue square: the centroid for group 3. Ellipses represent a 95% confidence level for the true location of the group’s mean. While the ellipse for Group 1 includes four observations, the ellipse in Group 3 does not include any observations of Group 3, since none of the contours in this group approached to the mean values significantly.

**Figure 9 plants-14-01522-f009:**
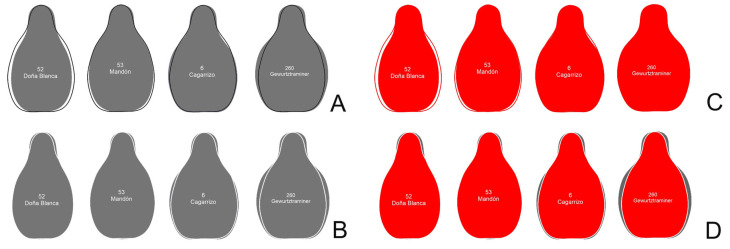
Schematic representation of the method to calculate *J*-index (percent of similarity of a set of images with a model). The model (average silhouette of Hebén) is superimposed here to contours of Doña Blanca, Mandón, Cagarrizo, and Gewürtztraminer, searching the maximum surface shared between the image and the model. In the left, above, the superimposed model is in black (**A**), and below, in white (**B**). When the images of the left are opened in Image J (right, red), the image on top (**C**) shows the total number of pixels in the figure (corresponding to the surface occupied by the model and seed image), while the image below (**D**) shows the shared surface (pixels shared by the image and the model).

**Table 1 plants-14-01522-t001:** Comparison of J-index values with two models: the average contour from the image program, and the Fourier model. Values are the means of 20 measurements (between brackets, coefficients of variation). Same superscript letter in a column indicates no significant difference.

Model	Hebén	Listán Prieto	Merseguera	Verdejo de Salamanca
Average contour (image program)	92.6 ^a^ (1.8)	92.0 ^a^ (2.2)	91.7 ^a^ (1.6)	92.0 ^a^ (0.8)
Average contour (Fourier)	93.2 ^a^ (1.8)	91.9 ^a^ (2.0)	92.4 ^a^ (1.2)	92.3 ^a^ (0.8)

**Table 2 plants-14-01522-t002:** Statistical comparison of the mean values of *J*-index with two models, aspect ratio (AR) and solidity (S), in Groups 2 and 3, of cultivars with *J*-index values superior to 94 with models Hebén and Chenin. The data are mean values (coefficients of variation between brackets). N = number of contours (1 contour per cultivar). Different superscript letters mean differences between files in the same column.

Groups	N	*J*-I Hebén	*J*-I Chenin	AR	S
		95.1 ^a^ (0.9)	93.8 ^a^ (1.4)	1.6 ^a^ (3.0)	970 ^a^ (0.7)
Group 2 (model Hebén)	60	93.7 ^b^ (0.9)	95.2 ^b^ (0.8)	1.7 ^b^ (2.6)	963 ^b^ (0.9)
Group 3 (model Chenín)	44	95.1 ^a^ (0.9)	93.8 ^a^ (1.4)	1.6 ^a^ (3.0)	970 ^a^ (0.7)

**Table 3 plants-14-01522-t003:** Statistical comparison of the curvature values (Max, Min, Mean, and Max to Mean ratio) in two morphotypes represented by the Hebén and Chenin models. The data are the mean values (coefficients of variation between brackets) of the maximum, minimum, and average curvature values of six representative contours of each type. N = number of contours (1 contour per cultivar). Different superscript letters mean differences between files in the same column.

Morphotype	N	AR	S	Max	Min	Mean	Ratio
Hebén	6	1.7 ^a^ (3.3)	975 ^a^ (0.3)	0.40 ^a^ (9.5)	−0.56 ^a^ (14.6)	0.08 ^a^ (19.6)	5.20 ^a^ (26.2)
Chenin	6	1.7 ^a^ (7.9)	956 ^b^ (0.5)	0.49 ^b^ (12.8)	−0.81 ^b^ (11.1)	0.07 ^a^ (17.9)	7.21 ^b^ (22.4)

**Table 4 plants-14-01522-t004:** A complete list of the species and cultivars used in the present study. U = unpublished.

N	Cultivar	Reference	N	Cultivar	Reference
1	Airén	[30,40,41]	137	Montonera	[37,41]
2	Alarije	[30,40,41]	138	Montúa	[40,41]
3	Albana	[37,41]	139	Montúa 2024	[40]
4	Albarín blanco	[41]	140	Moraté	[37,41]
5	Albariño	[41]	141	Moravia Agria	[41]
6	Albillo de Granada	[41]	142	Moravia dulce	[41]
7	Albillo del Pozo	[37,41]	143	Morenillo	[41]
8	Albillo Mayor	[41]	144	Morisca	[41]
9	Albillo Real	[30,41]	145	Moristel	[41]
10	Alcañón	[41]	146	Moscatel de Alejandría	[30,40]
11	Aledo	[41]	147	Moscatel de Angués	[40]
12	Aledo Real	[41]	148	Moscatel de Angués 2024	[40]
13	Alphonse Lavalleé	U	149	Moscatel de Grano Menudo	[30,37,41]
14	Arcos	[37,41]	150	Moscatel de Hamburgo	
15	Áurea	[37,41]	151	Moscatel Rosa	[41]
16	Azargón	[41]	152	Negra Rayada	[41]
17	Bastardo Blanco	[37,41]	153	Negreda	[37,40,41]
18	Bastardo Negro	[41]	154	Negreda 2024	[40]
19	Batista	[41]	155	Ohanes	[41]
20	Beba	[30,37,40,41]	156	Ondarrabi Beltza	[37,41]
21	Benedicto	[37,41]	157	Ondarrabi Zuri	[41]
22	Benedicto falso	[41]	158	Palomino Fino	[30,41]
23	Bermejuela	[41]	159	Palote	[41]
24	Blanquiliña	[41]	160	Pampolat de Sagunto	[41]
25	Bobal	[30,41]	161	Pardillo	[41]
26	Borba (Riesling Itálico)	U	162	Parduca	[41]
27	Brancellao	[41]	163	Parellada	[41]
28	Bruñal	[30,41]	164	Parraleta	[41]
29	Cabernet Franc	[41]	165	Pedro Ximénez	[30,40]
30	Cabernet Sauvignon	[41]	166	Pedro Ximénez 2024	[40]
31	Cadrete	[41]	167	Pedrol	[41]
32	Cagarrizo	[41]	168	Perruno	[41]
33	Caíño Blanco	[41]	169	Picapoll Blanco	[41]
34	Caíño Longo	[41]	170	Picapoll Tinto	[41]
35	Caiño Tinto	[30,41]	171	Pinot Noir	[41]
36	Callet	[41]	172	Pintada	[41]
37	Cañaroyo	[41]	173	Planta Fina	[40,41]
38	Cardinal	[41]	174	Planta Fina 2024	[40]
39	Cariñena Blanca	[41]	175	Planta Mula	[37,41]
40	Cariñena Roja	[41]	176	Planta Nova	[37,41]
41	Carrasquín	[41]	177	Prieto Picudo	[30,41]
42	Castañal	[41]	178	Puerto Alto	[41]
43	Castellana Blanca	[30,41]	179	Quiebratinajas Rosa	U
44	Casteloa	U	180	Quigat	[40,41]
45	Cayetana Blanca	[30,40,41]	181	Quigat 2024	[40]
46	Cayetana Blanca 2024	[40]	182	Ragol	[41]
47	Cenicienta	U	183	Ratiño	[41]
48	Chardonnay	[41]	184	Rayada Melonera	[37,41]
49	Chasselas Blanc	[41]	185	Red Globe	U
50	Chenin	U	186	Regina de Vignetti	U
51	Ciruela roja	U	187	Riesling	[41]
52	Coloraillo	U	188	Rocía	[41]
53	Corazón de Cabrito	[40]	189	Rojal tinta	[41]
54	Corchera	[37,41]	190	Rosetti	[41]
55	Crepa	[41]	191	Rubeliza	[41]
56	Cuatendrá	[41]	192	Rufete	[41]
57	De Cuerno	[30,41]	193	Rufete Serrano	[37,41]
58	Derechero	[41]	194	Sabaté	[41]
59	Diega	[37,41]	195	Sabro	[40,41]
60	Dominga	[30,41]	196	Sabro 2024	[40]
61	Don Mariano	U	197	Salvador	[41]
62	Doña Blanca	[30,41]	198	Sanguina	[37,41]
63	Doradilla	[41]	199	Sauvignon Blanc	[41]
64	Espadeiro	[41]	200	Semillon	U
65	Esperó de Gall	[41]	201	Señá	[40,41]
66	Estaladiña	[37,41]	202	Señá 2024	[40]
67	Excursach	[41]	203	Souson	[41]
68	Fernâo Pires	[41]	204	Sumoll	[40,41]
69	Ferral	[40,41]	205	Sumoll 2024	[40]
70	Ferrón	[41]	206	Sylvestris BA2.3	[41]
71	Fogoneu	[41]	207	Sylvestris CA13.3	[41]
72	Folle Blanche	[41]	208	Sylvestris CA13.4	[30,41]
73	Forastera	[41]	209	Sylvestris CA13.6	[30,41]
74	Forcallat Tinta	[40,41]	210	Sylvestris CA2.9	[41]
75	Gabriela	[41]	211	Sylvestris CA2.9b	[41]
76	Garnacha Blanca	[41]	212	Sylvestris CC1.5	[41]
77	Garnacha Peluda	[41]	213	Sylvestris CO4.7	[41]
78	Garnacha Roja	[41]	214	Sylvestris FR1.1	[41]
79	Garnacha Tinta	[30,41]	215	Sylvestris FR1.4	[41]
80	Garnacha Tintorera (Alicante Bouschet)	U	216	Sylvestris H6.1	[41]
81	Garrido Fino	[41]	217	Sylvestris H6.5	[41]
82	Garrido Macho	[41]	218	Sylvestris J1.4	[41]
83	Gewürztraminer	[30,41]	219	Sylvestris J2.4	[41]
84	Giro Negro	[37,41]	220	Sylvestris NA2.4b	[41]
85	Godello	[41]	221	Sylvestris NA3.2b	[41]
86	Gonfaus	[37,41]	222	Sylvestris S3.5b	[41]
87	Gorgollasa	[37,41]	223	Sylvestris SE2.1	[30,41]
88	Graciano	[30,41]	224	Sylvestris SE2.4	[41]
89	Gran Noir	[41]	225	Sylvestris SE2.6	[41]
90	Granadera	[41]	226	Sylvestris SE2.7	[41]
91	Gualarido	[41]	227	Syrah	[41]
92	Hebén	[30,40,41]	228	Tarragoní	[40]
93	Heben 2024	[40,41]	229	Tarragoní 2024	[40]
94	Imperial	[30,41]	230	Tempranillo	[30,40]
95	Italia (Moscatel Romano)	[41]	231	Terriza	[37,41]
96	Jaén rosado	[41]	232	Teta de Vaca	[30,41]
97	Jaén tinto	[41]	233	Tinto Bastardo	[41]
98	Jaén tinto 2024	[41]	234	Tinto de la Pampana Blanca	[41]
99	Jarrosuelto	[37,41]	235	Tinto de Navalcarnero	[41]
100	Jerónimo	[40,41]	236	Tinto Fragoso	[37,41]
101	Juan García	[30,41]	237	Tinto Jeromo	[37,41]
102	Juan García 2024	[40]	238	Tinto Velasco	[41]
103	Lado	[41]	239	Torralba	[40,41]
104	Legiruela	[41]	240	Torrontés	[41]
105	Listán del Condado	[41]	241	Tortosí	[41]
106	Listán Negro	[41]	242	Tortozón	[30,41]
107	Listán Prieto	[30,40,41]	243	Tortozona Tinta	[30,40,41]
108	Listán Prieto 2024	[40]	244	Tortozona Tinta 2024	[40]
109	Loureira	[41]	245	Treixadura	[41]
110	Lucomol	[41]	246	Trepat	[40,41]
111	Macabeo	[30,41]	247	Trepat 2024	[40]
112	Malbec	[41]	248	Trincadeira das Pratas- Allarén	[41]
113	Malvar	[40,41]	249	Trobat	[41]
114	Malvasía Aromática	[30,41]	250	Verdejo	[30,41]
115	Malvasía Volcánica	U	251	Verdejo de Salamanca	[40,41]
116	Mandón	[40]	252	Verdejo de Salamanca 2024	[40]
117	Mandregue	[37,41]	253	Verdil (Merseguera)	[41]
118	Mansés de Capdell	[41]	254	Vidadillo	[41]
119	Mansés de Tibbus	[41]	255	Vijariego Blanco	[41]
120	Manto negro	[41]	256	Viognier	[41]
121	Mantúo de pilas	[41]	257	*Vitis amurensis*	[28]
122	Marfal	[41]	258	*Vitis berlandieri*	[28]
123	Marfileña	[41]	259	*Vitis californica*	[28]
124	Maturana Blanca	[41]	260	*Vitis candicans*	[28]
125	Mazuela	[30,41]	261	*Vitis doaniana*	[28]
126	Mencía	[41]	262	*Vitis labrusca*	[28]
127	Merenzao	[41]	263	*Vitis riparia*	[28]
128	Merlot	[41]	264	*Vitis rupestris*	[28]
129	Merseguera 2020	[40]	265	Viura	[40,41]
130	Merseguera 2024	[40]	266	Viura 2024	[40]
131	Miguel de Arco	[40]	267	Xarello	[40,41]
132	Miguel de Arco 2024	[40]	268	Xarello 2024	[40]
133	Mollar Cano	[30,40]	269	Xarello Rosado	[41]
134	Mollar Cano 2024	[40]	270	Zalema	[30,41]
135	Monastrell	[30,41]	271	Zurieles	[41]
136	Mondragón	[41]			

## Data Availability

See the Appendix A section above.

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
