# Peer review of "Seed Morphometry Reveals Two Major Groups in Spanish Grapevine Cultivars"

_plants, 2025, doi:10.3390/plants14101522_

Round 1

Reviewer 1 Report

Comments and Suggestions for Authors

This manuscript, titled "Seed morphometry reveals two major groups in Spanish grapevine cultivars," investigates the morphological characteristics of seeds from a selection of Spanish grapevine cultivars. The study employs morphometric analysis to explore potential groupings and relationships within this important germplasm. While the topic of grapevine cultivar characterization is of significant interest to the readership of Plants, particularly given the economic and cultural importance of viticulture, this review will delve into the methodology, results, and interpretation presented by the authors to assess the manuscript's clarity, rigor, and contribution to the field. The following sections will critically evaluate the experimental design, the statistical analyses employed, the presentation of the findings, and the overall discussion of the results in the context of existing knowledge.

While the authors employed Elliptic Fourier analysis, a valuable technique in geometric morphometrics for quantifying landmark-poor outlines, and successfully derived harmonic coefficients, a significant oversight appears to be the lack of subsequent analysis using Principal Component Analysis. Although EFA provides a powerful means of obtaining quantitative shape variables, failing to subject these coefficients to PCA represents a missed opportunity. PCA, a statistical method known for its ability to reduce dimensionality while retaining crucial information, would have likely offered a more insightful avenue for revealing relationships between the studied groups. The authors' reliance on the J-index for assessing similarity, while potentially informative, is much more robust than the multivariate approach afforded by PCA in discerning subtle yet significant shape differences. In my view, this absence of PCA application to the EFA-derived coefficients is the most substantial drawback of this work, particularly considering the potential for PCA to provide a clearer understanding of the morphological distinctions between the groups under investigation. The integration of PCA would have undoubtedly strengthened the analysis and provided more compelling evidence regarding the significance of the observed shape variations.

It's a pity that the names of the species analysed were not provided within the main body of the article. This information is crucial enough that it should have been included in the article body.

The use of "sylvestris" is puzzling. What did the authors intend to convey? I presume they are not referring to the species Vitis sylvestris, as this name is synonymous with Vitis vinifera.

How should the term "son of" be interpreted? This is the first time I've encountered this term in a botanical text.

The use of the terms "seed species" (line 63) and "seed populations" (line 70) raises my concern.

What were the rationales behind constructing Models 1 and 2 with this specific set of varieties and years? Were any preliminary analyses conducted that provided a basis for these choices? How significant are the differences between Hebén 2020 and 2024, as well as between Chenin and Gewürztraminer, and how might this have influenced the subsequent results?

I find some sentences in the text to be rather unclear, for example: "all Vitis species other than V. vinifera cultivars and most sylvestris seeds." The phrase "most sylvestris seeds" is particularly vague. I must admit that the process of defining the working groups in this article is not entirely clearly described and could lead to doubts during later interpretation.

While reviewing the cited sources, I noticed a citation error in, for example, article number 46. Originally cited as: Dong, Yang, et al. "Dual domestications and origin of traits in grapevine evolution." Science, vol. 373, no. 6559, 3 September 2021, pp. 1085-1092, doi:10.1126/science.abg6115, the correct citation should be: Yang Dong et al., Dual domestications and origin of traits in grapevine evolution. Science 379, 892-901 (2023). DOI: 10.1126/science.add8655. The publication year, volume number, and DOI are all incorrect in the original citation. It is important to pay attention to accurate citations and carefully verify the entire list of references.

The authors use the term "Iberia" in opposition to "Western Europe," which raises some concerns. Spain, Portugal, and Andorra are often considered part of Western Europe. Given that the paper focuses on winemaking, it's also worth noting the additional risk of misunderstanding the term "Iberia," as it is also the ancient name of a region in Georgia, a significant centre for winemaking and often considered the cradle of viticulture. Therefore, does "Iberia" simply refer to the Iberian Peninsula? If so, it would be clearer to use that term. Furthermore, the definition of "Western Europe" remains somewhat ambiguous in this context. Does it include France and Germany, but also Italy? Clarification on these geographical terms would enhance the precision and avoid potential confusion for the reader.

In my opinion, the Results and Discussion sections are written somewhat chaotically and at times are too verbose, with the frequent listing of numerous variety names. This significantly hinders the reader's ability to grasp the core findings and the essence of the discussion. I believe these sections lack a clearer and more logical presentation. Perhaps a different approach to presenting the results, including alternative types of graphical representations, would be beneficial. I advocate for simplifying these sections and focusing on the most significant findings, avoiding an excessive accumulation of details (such as lists of variety names), so that the reader can more quickly understand the obtained results and their implications for the current state of knowledge.

I also question the comparison of maxs, mins, and means (Table 3). Wouldn't it be more appropriate to compare the statistical populations using a test that takes such values into account? A non-parametric equivalent of the Student's t-test should serve this function well.

Attention to technical consistency is needed throughout the manuscript, including proper formatting of units (e.g., “cm” instead of “cm.”). Such issues, while minor, reflect on the manuscript’s professionalism and editorial rigor.

In its current form, the manuscript contains substantial methodological and editorial deficiencies that compromise its clarity and scientific rigor. The main areas requiring revision include:

  • Inclusion of PCA in the morphometric analysis.
  • Clarification of experimental design and sample selection.
  • Greater precision in terminology and taxonomic description.
  • Improved organization and graphical presentation of results.
  • Correction of reference and formatting errors.

With significant revisions addressing these points, the manuscript could offer a valuable contribution to the field of grapevine morphometrics and cultivar characterization.

Author Response

Comments 1:

This manuscript, titled "Seed morphometry reveals two major groups in Spanish grapevine cultivars," investigates the morphological characteristics of seeds from a selection of Spanish grapevine cultivars. The study employs morphometric analysis to explore potential groupings and relationships within this important germplasm. While the topic of grapevine cultivar characterization is of significant interest to the readership of Plants, particularly given the economic and cultural importance of viticulture, this review will delve into the methodology, results, and interpretation presented by the authors to assess the manuscript's clarity, rigor, and contribution to the field. The following sections will critically evaluate the experimental design, the statistical analyses employed, the presentation of the findings, and the overall discussion of the results in the context of existing knowledge.

While the authors employed Elliptic Fourier analysis, a valuable technique in geometric morphometrics for quantifying landmark-poor outlines, and successfully derived harmonic coefficients, a significant oversight appears to be the lack of subsequent analysis using Principal Component Analysis. Although EFA provides a powerful means of obtaining quantitative shape variables, failing to subject these coefficients to PCA represents a missed opportunity. PCA, a statistical method known for its ability to reduce dimensionality while retaining crucial information, would have likely offered a more insightful avenue for revealing relationships between the studied groups. The authors' reliance on the J-index for assessing similarity, while potentially informative, is much more robust than the multivariate approach afforded by PCA in discerning subtle yet significant shape differences. In my view, this absence of PCA application to the EFA-derived coefficients is the most substantial drawback of this work, particularly considering the potential for PCA to provide a clearer understanding of the morphological distinctions between the groups under investigation. The integration of PCA would have undoubtedly strengthened the analysis and provided more compelling evidence regarding the significance of the observed shape variations.

Response 1 (PCA):

Following your advice, the PCA analysis is now included in a new subsection of the Results section (2.8. PCA defines three morphological types), including a new figure (Figure 8.  PCA analysis). The results of the PCA are also mentioned in the Introduction and Discussion sections.The species and cultivars presented here are those with lower similarity to the models and include all seeds belonging to different Vitis species other than V. vinifera.

Comments 2:

It's a pity that the names of the species analysed were not provided within the main body of the article. This information is crucial enough that it should have been included in the article body.

Response 2 (names of the species and cultivars analysed):

The names of the cultivars are now given in the main body of the article in the Materials and methods section, Table 4, Section 4.1. Plant Material.

Comments 3: 

The use of "sylvestris" is puzzling. What did the authors intend to convey? I presume they are not referring to the species Vitis sylvestris, as this name is synonymous with Vitis vinifera.

Response 3 (use of therm "sylvestris"):

The term "sylvestris" has been now explained in the Introduction, as follows:

"…from sylvestris plants, i.e. those plants maintained in the IMIDRA collection that proceed from plants that once grew in the wild."

And the description is expanded in the Materials and Methods section:

“The “sylvestris” samples are 21 female plants surveyed between 2003 and 2009 in natural populations of riparian forests in the provinces of Badajoz, Caceres, Cadiz, Huelva, Jaén, Seville, Cordoba, Navarra and the French Basque Country. These plants were selected for being female and therefore producing seeds and for being populations not contaminated with cultivated vines and for representing samples from the entire Iberian Peninsula.”

Comments 4: 

How should the term "son of" be interpreted? This is the first time I've encountered this term in a botanical text.

Response 4 (use of therm "son of" discarded):

We agree that the term is not used in the botanical or ampelographic texts. We have changed it for offspring or progeny through the text.

Comments 5: 

The use of the terms "seed species" (line 63) and "seed populations" (line 70) raises my concern.

Response 5 (terms "seed species" and "seed populations"modified ):

Concerning the expressions "seed species" (line 63) and "seed populations" (line 70) these have been changed to “the seeds of many species and cultivars”, and “seeds of diverse species and cultivars”, respectively.

Comments 6: 

What were the rationales behind constructing Models 1 and 2 with this specific set of varieties and years? Were any preliminary analyses conducted that provided a basis for these choices? How significant are the differences between Hebén 2020 and 2024, as well as between Chenin and Gewürztraminer, and how might this have influenced the subsequent results?

Response 6 (Models):

This aspect is now mentioned briefly in the introduction:

"For shape comparisons, two new models were defined based on the contours of morphologically different and reference cultivars. Model 1 was made with the average contour of the seeds of the cultivar Hebén (2020 and 2024; Hebén model, for morphotype 1) and model 2 with an average contour of the European varieties Chenin and Gewürtztraminer (Chenin model, for morphotype 2)."

And more expanded in section 2.3 of the results section (Two new models):

“Both models were designed on the basis of their morphological peculiarities (higher solidity in Hebén, lower in Chenin) as well as their potential representativeness for the Iberian and West European varieties”

Also the average contours of Hebén 2020 and 2024 have been superimposed to Model Hebén and those of Chenin and Gewürztraminer to Model Chenin in Fig. 2, and this has been indicated in the text (Sect. 2.3).

Comments 7: 

I find some sentences in the text to be rather unclear, for example: "all Vitis species other than V. vinifera cultivars and most sylvestris seeds." The phrase "most sylvestris seeds" is particularly vague. I must admit that the process of defining the working groups in this article is not entirely clearly described and could lead to doubts during later interpretation.

Response 7 (phrase "most sylvestris seeds"):

This may be more clear after our response to comments 3 above.

Comments 8: 

While reviewing the cited sources, I noticed a citation error in, for example, article number 46. Originally cited as: Dong, Yang, et al. "Dual domestications and origin of traits in grapevine evolution." Science, vol. 373, no. 6559, 3 September 2021, pp. 1085-1092, doi:10.1126/science.abg6115, the correct citation should be: Yang Dong et al., Dual domestications and origin of traits in grapevine evolution. Science 379, 892-901 (2023). DOI: 10.1126/science.add8655. The publication year, volume number, and DOI are all incorrect in the original citation. It is important to pay attention to accurate citations and carefully verify the entire list of references.

Response 8 (References):

The references have been reviewed and ref. number 46 in the actual version has been obtained directly from PubMed.

Comments 9: 

The authors use the term "Iberia" in opposition to "Western Europe," which raises some concerns. Spain, Portugal, and Andorra are often considered part of Western Europe. Given that the paper focuses on winemaking, it's also worth noting the additional risk of misunderstanding the term "Iberia," as it is also the ancient name of a region in Georgia, a significant centre for winemaking and often considered the cradle of viticulture. Therefore, does "Iberia" simply refer to the Iberian Peninsula? If so, it would be clearer to use that term. Furthermore, the definition of "Western Europe" remains somewhat ambiguous in this context. Does it include France and Germany, but also Italy? Clarification on these geographical terms would enhance the precision and avoid potential confusion for the reader.

Response 9 (geographical regions):

Concerning the two geographical regions, the terms the Iberian Peninsula and Western Europe are now used through the article indicating two different regions: the Iberian Peninsula and other geographic locations in Western Europe, mainly France and Germany.

Comments 10: 

In my opinion, the Results and Discussion sections are written somewhat chaotically and at times are too verbose, with the frequent listing of numerous variety names. This significantly hinders the reader's ability to grasp the core findings and the essence of the discussion. I believe these sections lack a clearer and more logical presentation. Perhaps a different approach to presenting the results, including alternative types of graphical representations, would be beneficial. I advocate for simplifying these sections and focusing on the most significant findings, avoiding an excessive accumulation of details (such as lists of variety names), so that the reader can more quickly understand the obtained results and their implications for the current state of knowledge.

Response 10:

Figure 2 has been has modified and Figure 8 has been added. Also annotations have been added to the figure legends and the text has been simplified in some instances. We hope this may contribute to improve the readability. We agree that the subject is complex, but only giving a list of variety names associated with each morphological type can help to understand the basis of the relationship between multiple factors determining seed shape in the complex Vitis system .

Comments 11: 

I also question the comparison of maxs, mins, and means (Table 3). Wouldn't it be more appropriate to compare the statistical populations using a test that takes such values into account? A non-parametric equivalent of the Student's t-test should serve this function well.

Response 11:

The populations were compared with Mann-Whitney U test as indicated in section 4.7. Statistical Analysis. The Average contours (Acs) are treated as geometric curves and the measurements of curvature along each curve includes the max, min, and mean valuess. These values are statistically compared on Table 3.

Comments 12: 

Attention to technical consistency is needed throughout the manuscript, including proper formatting of units (e.g., “cm” instead of “cm.”). Such issues, while minor, reflect on the manuscript’s professionalism and editorial rigor.

Response 12:

Unit formatting has been revised and corrected throughout the text.

Reviewer 2 Report

Comments and Suggestions for Authors

“Seed morphometry reveals two major groups in Spanish grapevine cultivars€€.” is a good research topic. In this study, seed shape was analysed in a total of 271 cultivars and species of itis. After a general morphometric analysis including shape measurements, eight Fourier coefficients were obtained with Momocs for samples of 20-30 seeds representing each of the 271 cultivars analysed, and the average contour was plotted for each cultivar. Seed morphometric analysis, as used in this study, provides a useful approach for cultivar identification and characterisation, as well as for parental relationship analysis. The objectives of this study are clear, the methods are rigorous, and the data are comprehensive. However, there are some important points that need to be improved before further processing. 

In Keywords of the manuscript:

The keywords are too numerous, and it would be better to refine them to five.

In Abstract of of the manuscript:

The abstract of the manuscript includes too much description of the methods, taking up a large amount of space. In the final sentence of the abstract, the purpose and significance of the study should be highlighted.

In Introduction of the manuscript:

Line 52, Line 56, etc, the citation format in the text is incorrect; for example, [9,10,11,12,13,14] should be written as [9-14].

Line 80, 90, 91, the manuscript mentioned “………a total of 271 cultivars and species of Vitis………”, but it does not provide a detailed explanation of the representativeness of these samples in the context of the entire Spanish grapevine cultivars. This information should be supplemented.

The specific objectives and scientific significance of the study should be clearly stated in the final sentence of the introduction.

In Materials and Methods of the manuscript:

The manuscript provided a detailed introduction to the image analysis and morphometric methods. It is suggested to supplement more technical details, such as specific parameters of the image processing software and camera settings.

The resolution of some figures in the text could be appropriately increased, and their readability could be enhanced by adding clear annotations, such as for Figure 10.

In Discussion of the manuscript:

A discussion on the genetic mechanisms underlying the observed seed morphological traits could enhance the depth and breadth of the study.

In the References of the manuscript:

The writing format of the references needs to be carefully standardized. For example, the publication year in some references is in bold font, while in others it is not, such as Ref. 4 and Ref. 47. Some references contain extra parentheses. For example, Ref. 9.

Author Response

Comments 1:

In Keywords of the manuscript:

The keywords are too numerous, and it would be better to refine them to five.

Response 1:

Keyword number has been reduced to five.

Comments 2:

In Abstract of of the manuscript:

The abstract of the manuscript includes too much description of the methods, taking up a large amount of space. In the final sentence of the abstract, the purpose and significance of the study should be highlighted.

Response 2:

The description of the Methods has been reduced and the significance of the results has been highlighted in the last section of the Abstract.

Comments 3:

In Introduction of the manuscript:

Line 52, Line 56, etc, the citation format in the text is incorrect; for example, [9,10,11,12,13,14] should be written as [9-14].

Line 80, 90, 91, the manuscript mentioned “………a total of 271 cultivars and species of Vitis………”, but it does not provide a detailed explanation of the representativeness of these samples in the context of the entire Spanish grapevine cultivars. This information should be supplemented.

The specific objectives and scientific significance of the study should be clearly stated in the final sentence of the introduction.

Response 3:

Citation format has been corrected, and an explanation of the representativeness of the samples used in the context of the Spanish grapevine cultivars has been given.

The specific objectives and scientific significance of the study have been stated in the final sentences of the introduction.

Comments 4:

In Materials and Methods of the manuscript:

The manuscript provided a detailed introduction to the image analysis and morphometric methods. It is suggested to supplement more technical details, such as specific parameters of the image processing software and camera settings.

The resolution of some figures in the text could be appropriately increased, and their readability could be enhanced by adding clear annotations, such as for Figure 10.

Response 4:

More technical details have been added to the Materials and Methods section.

Some figures have been modified or added and annotations have been added to the figure legends to improve their readability.

Comments 5:

In Discussion of the manuscript:

A discussion on the genetic mechanisms underlying the observed seed morphological traits could enhance the depth and breadth of the study.

Response 5:

This aspect is mentioned at the end of the Discussion section:

The combination of quantitative measurements of shape, both traditional (aspect ratio, solidity) and others more recently introduced (J-index, curvature), will contribute to the definition of seed morphotypes which is a prerequisite for the identification of genetic factors controlling seed shape. Recent developments in association mapping open new opportunities to explore the genomic regions associated with seed morphotypes, a work based on the identification of phenotypes and the accurate selection of parents in crosses [46].

Comments 6:

In the References of the manuscript:

The writing format of the references needs to be carefully standardized. For example, the publication year in some references is in bold font, while in others it is not, such as Ref. 4 and Ref. 47. Some references contain extra parentheses. For example, Ref. 9

Response 6:

The format of the references has been reviewed and standarized according to the journal's instructions.

Round 2

Reviewer 1 Report

Comments and Suggestions for Authors

I appreciate that the revised version of the manuscript shows substantial improvement compared to the original submission. I would also like to thank the authors for their thorough and constructive responses. After reviewing the updated manuscript, I have a few technical comments and suggestions for further clarification.

I commend the authors for including the PCA analysis in the revised version. In lines 292–298, the contribution of variables to PC1 and PC2 is described, with references to codes such as A7, D2, D5, etc. However, it is not immediately clear what these labels correspond to. If this information is provided in the supplementary materials, it would be advisable to include a brief explanation or a reference in the main text. Key information that is essential for interpreting the results should be accessible without requiring the reader to consult supplementary files.

In Figure 8, ellipses are drawn around the centroids of the groups, but the statistical meaning of these ellipses is not explained. Typically, such ellipses represent a certain confidence level (e.g., 50%, 90%, or 95%), indicating the proportion of observations they encompass. In this case, the purpose of the ellipses remains unclear—for instance, the ellipse for group 3 does not include any of the corresponding data points, while the ellipse for group 1 includes four observations. This should be clarified either in the figure legend or in the Results section.

Lastly, I suggest that the passage in lines 104–108 be relocated from the Introduction to the Discussion section, as it presents findings derived from the authors’ own research rather than background or contextual information.

Author Response

Comments 1: 

In lines 292–298, the contribution of variables to PC1 and PC2 is described, with references to codes such as A7, D2, D5, etc. However, it is not immediately clear what these labels correspond to. If this information is provided in the supplementary materials, it would be advisable to include a brief explanation or a reference in the main text. Key information that is essential for interpreting the results should be accessible without requiring the reader to consult supplementary files.

Response 1: The following explanation has now been given in the text (Results section 2.8):

"PCA is based on the coefficients of Parametric Fourier equations: A1 to A8, and B1 to B8 are, respectively, the cosine and sine coefficients for the first eight harmonics corresponding to the x-coordinate. C1 to C8, and D1 to D8 are, respectively, the cosine and sine coefficients for the first eight harmonics corresponding to the y-coordinate."

Comments 2: 

In Figure 8, ellipses are drawn around the centroids of the groups, but the statistical meaning of these ellipses is not explained. Typically, such ellipses represent a certain confidence level (e.g., 50%, 90%, or 95%), indicating the proportion of observations they encompass. In this case, the purpose of the ellipses remains unclear—for instance, the ellipse for group 3 does not include any of the corresponding data points, while the ellipse for group 1 includes four observations. This should be clarified either in the figure legend or in the Results section.

Response 2: The following explanation has now been given in the legend to Figure 8 (Results section 2.8):

Ellipses represent a 95% of confidence level for the true location of the group’s mean. While the ellipse for group 1 includes four observations, the ellipse in group 3 does not include any observations of Group 3, due that none of the contours in this group approach to the mean values significantly.

Comments 3: Lastly, I suggest that the passage in lines 104–108 be relocated from the Introduction to the Discussion section, as it presents findings derived from the authors’ own research rather than background or contextual information.

This passage has been moved to the Discussion Section in the Article (Lines 426-430).

Thank you very much for all your comments. On behalf of the authors,

Emilio Cervantes (Corresponding author)